# Optimal Design and Analysis of 4.7 μm Hybrid Deep Dielectric High Efficiency Transmission Gratings

**DOI:** 10.3390/mi13101706

**Published:** 2022-10-10

**Authors:** Ye Wang, Xiuhua Fu, Yongyi Chen, Hangyu Peng, Li Qin, Yongqiang Ning, Lijun Wang

**Affiliations:** 1School of Opto-Electronic Engineering, Changchun University of Science and Technology, Changchun 130022, China; 2Changchun Institute of Optics, Fine Mechanics and Physics, Chinese Academy of Sciences, Changchun 130033, China; 3Zhongshan Institute of Changchun University of Science and Technology, Zhongshan 528437, China; 4Jlight Semiconductor Technology Co., Ltd., No. 1588, Changde Road, ETDZ, Changchun 130102, China; 5Peng Cheng Laboratory, No. 2, Xingke 1st Street, Nanshan, Shenzhen 518000, China; 6Academician Team Innovation Center of Hainan Province, Key Laboratory of Laser Technology and Optoelectronic Functional Materials of Hainan Province, School of Physics and Electronic Engineering, Hainan Normal University, Haikou 570206, China

**Keywords:** mid-infrared, diffraction grating, multilayer dielectric, transmission grating, high diffraction efficiency

## Abstract

There is currently no transmission grating with good diffraction efficiency in the 4.7 μm band. Metal gratings at this wavelength are all reflective gratings which has a diffraction efficiency of lower than 90% and lower laser damage threshold. In this paper, we bring up a design of a multi-layer transmission grating with both high diffraction efficiency and wide working wavelength band. We have proved that the transmission grating made of composite materials has an average diffraction effectiveness of more than 96% throughout the whole spectral range of 200 nm. Meanwhile, the theoretically computed transmission grating has a highest first-order diffraction efficiency of more than 99.77% at 4746 nm. This multilayer dielectric film transmission grating’s optimized design may further boost spectral beam combining power, providing a practical technique for increasing SBC power and brightness.

## 1. Introduction

The 4.7 μm transmission gratings have received increasing attention due to their potential applications. High-power mid-infrared systems working at 4.7 μm are needed in spectral beam combining systems, spectral detection systems, and wavelength modulation systems [1]. The diffraction grating is the key element for spectrum beam combining in order to realize the high-power high beam quality laser system [2]. The diffraction efficiency (DE) of the transmission grating in the high-power spectral beam combining system directly affects the usable energy of the entire system [3].

Our gratings are employed in the merging of high-energy laser beams. The laser damage threshold decreases when there is absorption in the dielectric layer, as a result, we select a non-absorbing dielectric layer. Furthermore, the non-absorbing dielectric layer has a substantially greater damage threshold than the present metal reflection grating [4]. The laser output is diffracted four times in the spectral beam combining (SBC) system, and the comprehensive diffraction efficiency is η^4^, where d is the value of diffraction efficiency; when η = 90%, η^4^ = 65.61%; and when η = 99%, η^4^ = 96.06%. It can be seen that the increase in η can largely increase the final usable power. As a result, increasing the grating’s DE from 90 [5] for ordinary metal grating to 99 percent will ultimately increase the comprehensive diffraction efficiency by 30%. Moreover, the transmission grating is better than the reflective grating because the optical elements are easier for adjusting. Moreover, there are very few reports about gratings working at the wavelength of 4.7 µm. To our best knowledge, there is no report about the multilayer transmission diffractive grating (MTDG) at this wavelength. The rectangular groove grating’s -1st order diffraction efficiency will not surpass 97 percent due to Fresnel reflection on the grating surface, and Fresnel reflection is caused by differences in the refractive index of materials. The research presented in this study focuses on buried transmission gratings with excellent diffraction efficiency and a high damage resistance threshold. Table 1 shows the parameters utilized in this simulation.

Metal gratings have lower laser damage threshold thus not suitable for high power laser application. Meanwhile the metal gratings have a limited diffraction of lower than 90%. There is currently no 4.7 μm transmission grating. In this paper, we bring up a design of a multi-layer transmission grating, including three anti-reflection films and a hybrid material grating structure due to the limited substrate selection. Our design has both highest diffraction efficiency of more than 99.77% at 4746 nm, and wide working wavelength band of 200 nm, from 4600 nm to 4800 nm, with the lowest diffraction effectiveness of more than 96%. This multilayer dielectric film transmission grating’s optimized design may further boost spectral beam combining power, providing a practical technique for increasing SBC power and brightness.

## 2. Optimal Multilayer Dielectric DBR Design in Grating Grooves

Figure 1 depicts a schematic illustration of the structure of the preliminary planned mid-infrared transmission grating. The grating layer structure and anti-reflection coating structure of the multi-layer dielectric film transmission grating are designed separately. The grating layer structure is used to achieve high DE within a wide spectrum range of the diffracted light, while the anti-reflection coating structure is utilized to annihilate reflection brought by the refractive index difference between the grating material and air, between the bottom SiO_2_ and the Si substrate, and between the Si substrate and air.

Period 4. 7 μm, the line density is 333.33 line/mm, accordingly the incidence angle is 51.5667°, and DE is greater than 96 percent within the width of 0.2 μm. The shape of the grating is set to be rectangular groove. These parameters are listed as Table 2 below:

In the mid-infrared range, the substrate and the dielectric film materials are severely restricted. The grating substrate in this study is made of double-polished Si. The intrinsic absorption of Si is very weak from 3 μm to 5 μm range [11]; Si, Ge, and SiO_2_ are the dielectric film materials, owing to their stable chemical characteristics, low loss in the mid-infrared band, having a high and low refractive index contrast, which can effectively minimize manufacturing difficulties. SiO_2_ is a low refractive index material with great firmness and damage resistance and a good fit with Si and Ge [12]. The optical film theory [13] calculates the refractive index of the film material at 4.7 μm based on deposition tests of Si, Ge, and SiO_2_ single-layer films to be n(Si) = 2.92, n(Ge) = 3.88, and n(SiO_2_) = 1.40.

It is to be motioned that at the interface of the substrate and air, there is another material–air interface that needs the anti-reflection coating to annihilate the reflection.

To enhance the DE of −1 order transmission, manufacturing error produced by the manufacturing process must be taken into account, and the chosen grating structure must have a wider tolerance of process parameters. We are mostly concerned with rectangular gratings in this section. As a result, the parameters to be tuned are groove depth and duty cycle, which is defined as the line width to grating period ratio.

Our structure includes three anti-reflection films (AR1, AR2, and AR3) and also a grating using hybrid materials, as well as two phase match layers (Ph1 and Ph2) as shown in Figure 1. AR1 is used to eliminate the reflection from the air to the grating structure. AR2 is used to annihilate the reflection from the grating to the substrate material. AR3 is used to reduce the reflection from the substrate to the air. The phase match layers Ph1 and Ph2 are used to match the phase, increase the largest DE, and adjust the peak DE wavelength. The grating layer is made up of multilayer materials, in order to reach high DE and large working wavelength with high DE. In Section 2.1 we simulated transmission grating based on pure Si grating, in Section 2.2 we simulated using hybrid materials instead of pure Si for the grating, and in Section 2.3 we simulated AR coatings, finally in Section 2.4 we simulated phase match layers, and we give the total transmission grating’s structure and simulated the DE and gives the electric field intensity.

### 2.1. Transmission Grating Based on Pure Si Grating

Consider the pure silicon grating buried in SiO_2_ material first; Figure 2 shows a schematic illustration of the pure silicon transmission grating structure. The grooves are filled with SiO_2_, which could be processed by atom layer deposition (ALD) method as demonstrated by [14].

The initial difficulty is to determine the SiO_2_ phase matching layer such that when DE achieves a maximum value at the center wavelength of 4.7 μm, the Si grating layer may reach a minimum value, which is advantageous for manufacture. 

When the aspect ratio is less than 1:1 for pure silicon gratings, from simulation, we can see that it is difficult to achieve very high diffraction efficiency of the grating in the spectral range of 4.6–4.8 μm; Figure 3 shows that when the thickness of the pure silicon grating layer is 3 μm, the diffraction grating has the highest diffraction efficiency, and the diffraction efficiency is 89.9%. The minimum DE from 4.6 μm to 4.8 μm reached 68.85%. The difference between the maximum and the minimum DE reaches 21.05%. The diffraction effectiveness of the grating rapidly drops when the thickness of the pure silicon grating layer is less than or larger than this thickness.

### 2.2. Using Hybrid Materials Instead of Pure Si for the Grating

The total DE value for pure silicon gratings is less than 90%, which is far from sufficient for high-power spectrum beam combining. In order to optimize both the highest DE and the total usable spectrum range, we build the grating using hybrid materials, which is composed of Si and SiO_2_ materials, such that the hybrid material may modify the effective refractive index of the grating, resulting in a high DE and a wide working spectrum range. The total hybrid grating is also buried in SiO_2_ material, as shown in Figure 4 below.

The structural characteristics of the more sophisticated mixed-material grating layer in the 4.7 μm transmission grating structure are analyzed using Comsol Multiphysics simulation software. We use Si/SiO_2_ DBR materials for the grating, and we use SiO_2_ to fill the grating grooves. The DBR has 6.5 pairs, with Si at both ends. The grating made of DBR materials can be considered an approximation of mixed material. The varied material thicknesses, Si and SiO_2_, give the overall thickness of the grating. By altering the silicon to oxygen ratio, it is comparable to altering both the grating thickness and the overall effective refractive index at the same time. Figure 5 shows that with a given incidence angle, we simulated the connection between the incident wavelength and the diffraction efficiency when the thickness of SiO_2_ ranges from 0.615 to 0.635 μm and the thickness of Si ranges from 0.13 to 0.15 μm, the incident angle value is 51.5667°. In all simulations, the polarization direction is perpendicular to the paper plane. When the thickness of Si is altered, it can be observed that there are always certain options that fulfill greater DE requirements, which means larger processing tolerances. Peak diffraction efficiency is 99.95 percent when the thickness of Si is 0.140 μm. When the thickness of Si decreases, the overall spectral curve shifts to blue, and the peak diffraction efficiency falls by a small amount; when the thickness of Si grows, the overall spectral curve shifts to red, and the peak diffraction efficiency decreases to some extent. The peak diffraction efficiency also decreases. To achieve the best diffraction efficiency in the specified wavelength band, the suitable thickness and ratio of SiO_2_ and Si must be chosen.

The blue shift of the complete spectral curve is achieved under different incidence angles when the thickness of SiO_2_ is 0.625 μm and the thickness of Si is 0.130 μm, as shown in Figure 6a; when the thickness of SiO_2_ is 0.625 μm and the thickness of Si is 0.140 μm, different incident angles are obtained. When the incidence angle is 51°, the peak diffraction efficiency at the center wavelength of 4.7 μm is 99.99 percent, and the diffraction efficiency in the bandwidth of 0.2 μm is greater than 96 percent, as shown in Figure 6b. The red shift of the complete spectral curve is achieved under different incidence angles when the thickness of SiO_2_ is 0.625 μm and the thickness of Si is 0.150 μm, as shown in Figure 6c.

The operating spectrum range (minimum DE > 96 percent) of the high-power spectral combiner may reach more than 200 nm. When the thickness of Si is 0.140 μm and the thickness of SiO_2_ is 0.625 μm, as shown in Figure 7a. This DE is also greater than that of metal gratings, and the damage threshold of high-power lasers will be larger since absorption losses are eliminated. Figure 7b depicts the simulated electric field distribution.

#### The Effect of Duty Cycle

The wavelength of the incident light is 4.7 μm, and the incidence angle is 51.5667°. The grating’s groove depth and duty cycle are investigated. There is an inherent manufacturing fault in the groove width during the actual production process, which mostly influences the filling factor and hence the DE. The question is how significant the effect is and if it can be tolerated. We increased the thickness of Si to 0.140 μm and the thickness of SiO_2_ to 0.625 μm. Figure 8 depicts simulations for various fill factors (ff) at various incidence angles and wavelengths.

The simulation results demonstrate that when the filling factor is less than 0.5, there is always a suitable incidence angle for the overall spectral DE larger than 96 percent. When the fill factor exceeds 0.5, the minimum DE falls dramatically to 4.6 μm. The DE is hardly more than 95 percent at wavelengths shorter than 4.7 μm. These simulation results demonstrate that our design solutions have a reasonably broad process tolerance, which can accommodate parameter changes caused by overexposure or ICP etching.

When the grating duty cycle is between 0.45 and 0.55, the peak diffraction efficiency of the grating in the spectral region of 4.8 μm is around 98 percent, providing a suitable guidance for grating production and a broad process tolerance, as shown in Figure 9.

In comparison to the pure Si grating, refractive hybrid grating has a maximum grating DE of 99.95%, and the minimum DE reached 95.36%, the diffraction efficiency of the mixed material grating is greater than 96 percent in the spectral range of 4.6–4.8 μm.

It is a significant edge that hybrid material grating can improve both the highest DE and the overall DE larger than 96% for more than 200 nm.

### 2.3. AR Coating Design

Reflection at the air-substrate border can be decreased by using AR coating technology [15]. One of the most basic AR coatings is created by combining one or more pairs of dielectric coatings with high and low refractive indices. The physical thickness of the high or low refractive index dielectric coating may be adjusted to reduce reflected waves. This AR coating method is exclusively applicable to transmission gratings.

The substrate material for the 4.7 μm transmission grating is Si, while the dielectric film materials are Si, Ge, and SiO_2_. Ge, a high refractive index material, is more resistant to laser damage than Si. Due to its intrinsic material features, the low refractive index material SiO_2_ has a high anti-laser damage threshold [16], and the absorption coefficient of the thin film created in the mid-infrared band is minimal. The absorption of the film itself is neglected when estimating optical performance in this working band, and the dispersion phenomena of the refractive index changing with wavelength is not taken into account. The anti-reflection coating is built using the professional thin film design program OptiLayer, such that the diffraction efficiency of TE and TM is as near to 1 in the wavelength range under the incidence of the matching Littrow angle at the center wavelength as feasible. Set the initial film system environment to 4.7 μm for the incident wavelength and 333.33 lines/mm for the number of grating lines. As a result, the incidence angle, or Littrow angle, is 51.5667°, the incident medium is air. Because antireflection coatings are often created in an environment including electromagnetic and plasma fields, the photoelectrode value technique cannot be utilized to monitor film thickness. As a result, using a non-regular technique for AR coating design will provide more design and manufacturing freedom.

High and low refractive index conversion layers of various thicknesses can be used to provide high transmittance in the mid-infrared region, according to the principle of optical thin films. One layer in the film system of the optical film can be expressed as
(1)in cosδ1iη1sinδ1iη1sinδ1cosδ1

The film’s useful parameters are all contained in what is known as the characteristic matrix of the film. From the substrate and film’s characteristic matrix, Y = *C*/*B*, one may derive the combined admittance.

Similar results can be derived for the multilayer film in terms of the characteristic matrix of the film system:(2)$$\left(\begin{array}{c} B\\ C \end{array}\right)=\prod_{j=1}^{k}\left(\begin{array}{cc} \cos\delta_{1} & \frac{i}{\eta_{1}}\sin\delta_{1}\\ i\eta_{1}\sin\delta_{1} & \cos\delta_{1} \end{array}\right)\left(\begin{array}{c} 1\\ \eta_{k+1} \end{array}\right)$$
(3)$$\delta_{j}=\frac{2\pi }{\lambda }N_{j}d_{j}\cos\theta_{j}$$
$$\mathrm{in}\;\left(\begin{array}{cc} \cos\delta_{1} & \frac{i}{\eta_{1}}\sin\delta_{1}\\ i\eta_{1}\sin\delta_{1} & \cos\delta_{1} \end{array}\right)$$
is the feature matrix in the *j*th layer, and where δ is the phase, *λ* is the incident wavelength, *θ* is the incident angle, and *η* is the diffraction efficiency value. The transmittance and absorption of the film system can be calculated by Equation (3) [17].

Different requirements can be met for non-regular film systems through intricate calculations in accordance with the formula derived from the aforementioned optical film theory, when combined with mathematical operations and the potent optimization function offered by the film system design software. In order to decrease technical difficulty and the cumulative impact of mistakes made during the production of the film system, it is important to pay attention to the minimal number of film layers when developing the film system. Last but not least, Table 3 displays the optimum film design outcomes computed by Optilayer software.

Figure 10 depicts the antireflection coating AR1, AR2, and AR3’s theoretical design spectral curve for the structure of Figure 1. The figure shows that the spectral transmittance at the working wavelength of 4.7 μm is the greatest to match the design and application criteria of the transmission grating in the mid-infrared region. The theoretical transmittance is 99.99 percent in the AR1 film structure; the theoretical transmittance is 99.98 percent in the AR2 film structure; the theoretical transmittance is 98.8 percent in the AR3 film structure, and the average spectral transmittance is more than 98.5 percent in the 0.200 μm bandwidth.

In order to perform an optimization analysis, the planned anti-reflection coating structure was introduced to the Comsol Multiphysics simulation program. The resulting transmittance spectrum of the overall grating structure is displayed in Figure 10. The grating layer structure and the anti-reflection coating were perfectly suited.

### 2.4. Phase Match Layer

In this part, we consider both the grating designed in 2.2 and the AR coating designed in 2.3. These parts are arranged as shown in Figure 1, the grating layers are separated by two layers of SiO_2_. The bottom SiO_2_ serves mainly as the phase match layer, and the top SiO_2_ needs to be sufficiently thick so as to be counted as body material instead of being counted as part of the grating layer.

The top phase matching layer 1 thickness is simulated from 2500 nm to 3100 nm, with a 100 nm interval. Figure 11a shows that as the thickness of the phase matching layer 1 grows, so does the diffraction efficiency. When the thickness reaches 2800 nm, the diffraction efficiency reaches its maximum; when the thickness exceeds 2800 nm, the diffraction efficiency remains constant. The phase matching layer 2’s role is to provide phase matching between the entire grating and the multilayer dielectric anti-reflection coating, determining whether the highest diffraction efficiency is at the center wavelength of 4700 nm, and different thicknesses influence the highest value of the diffraction efficiency.

The grating field intensity distribution for the whole structure, including the substrate, Multilayer Dielectric AR Coatings and Reflective Coatings, Phase matching SiO_2_-1 and SiO_2_-2, grating layer and air layer is shown in Figure 12.

## 3. Conclusions

Owning to the metal grating’s absorption, the diffraction efficiency and laser damage threshold of the grating are very low, and the metal gratings in the 4.7 μm band are all reflective gratings. We demonstrated a transmission grating in the 4.7 μm band. Transmission grating structure includes three anti-reflection films (AR1, AR2, and AR3) and also a grating using hybrid materials, as well as two phase match layers (Ph1 and Ph2). According to our simulation, this grating has both the highest diffraction efficiency of more than 99.77% at 4746 nm, and wide working wavelength band of 200 nm, from 4600 nm to 4800 nm, with the lowest diffraction effectiveness of more than 96% (with period 3 μm, incident angle 51.5667°, filling factor 0.5). This multilayer dielectric film transmission grating’s optimized design may further boost spectral beam combining power, providing a practical technique for increasing SBC power and brightness. Future research will concentrate on the manufacture of gratings, experimental testing, and application testing of high-power laser beam combining systems.

## Figures and Tables

**Figure 1 micromachines-13-01706-f001:**
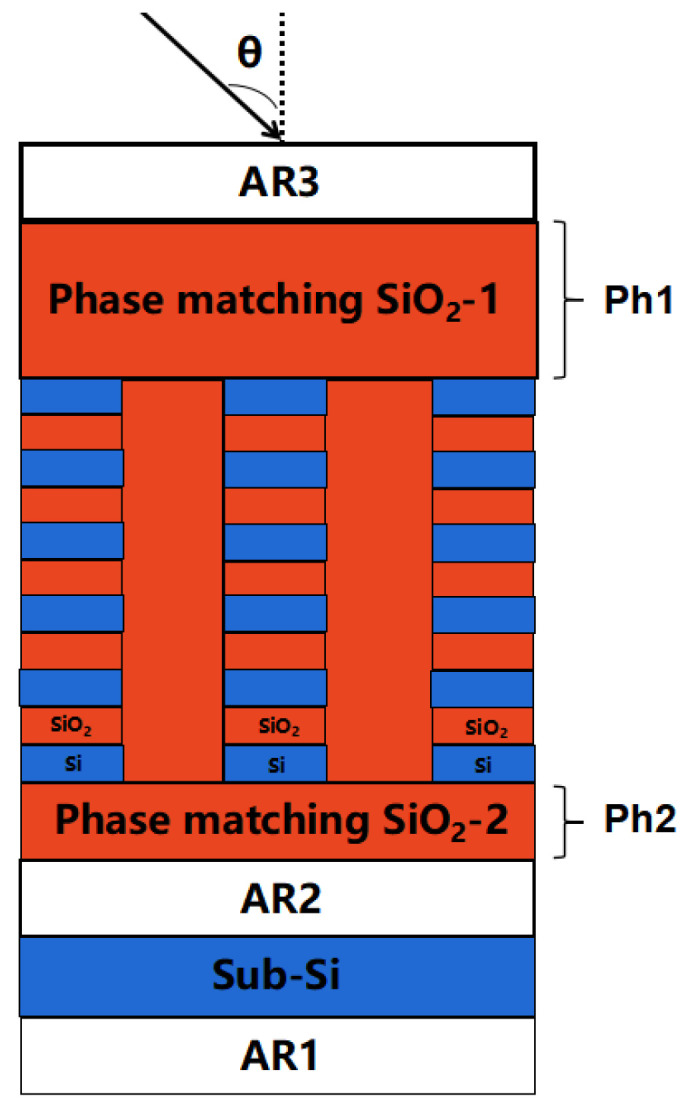
Schematic diagram of 4.7 μm transmission grating structure.

**Figure 2 micromachines-13-01706-f002:**
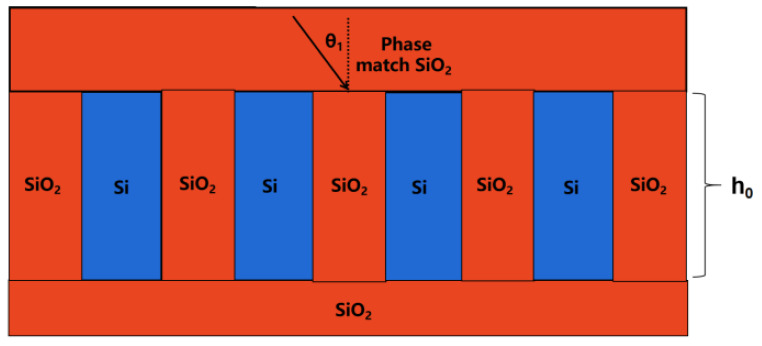
Schematic diagram of mid-infrared pure Si transmission grating structure.

**Figure 3 micromachines-13-01706-f003:**
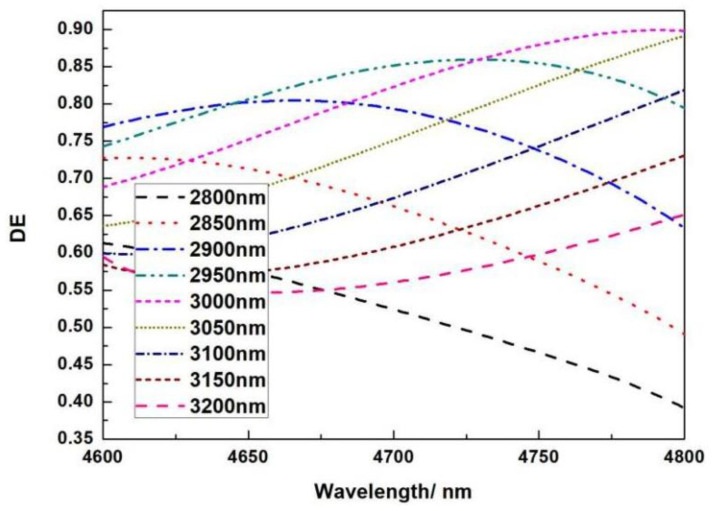
The relationship between wavelength and DE for pure silicon grating with different thicknesses h_0_.

**Figure 4 micromachines-13-01706-f004:**
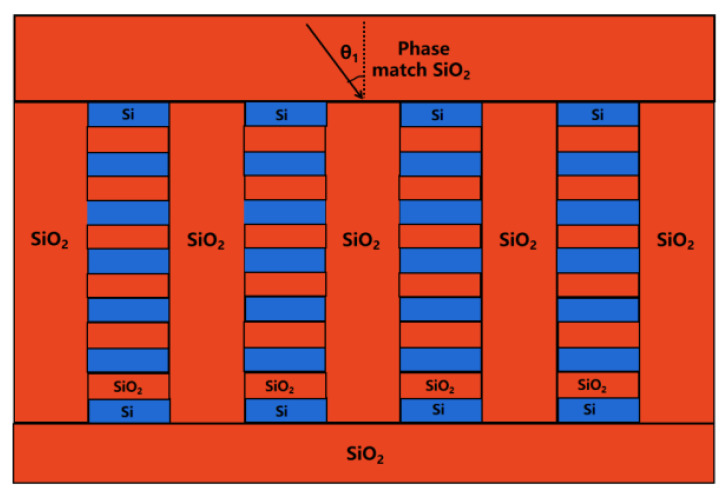
Hybrid grating buried in SiO_2_ material.

**Figure 5 micromachines-13-01706-f005:**
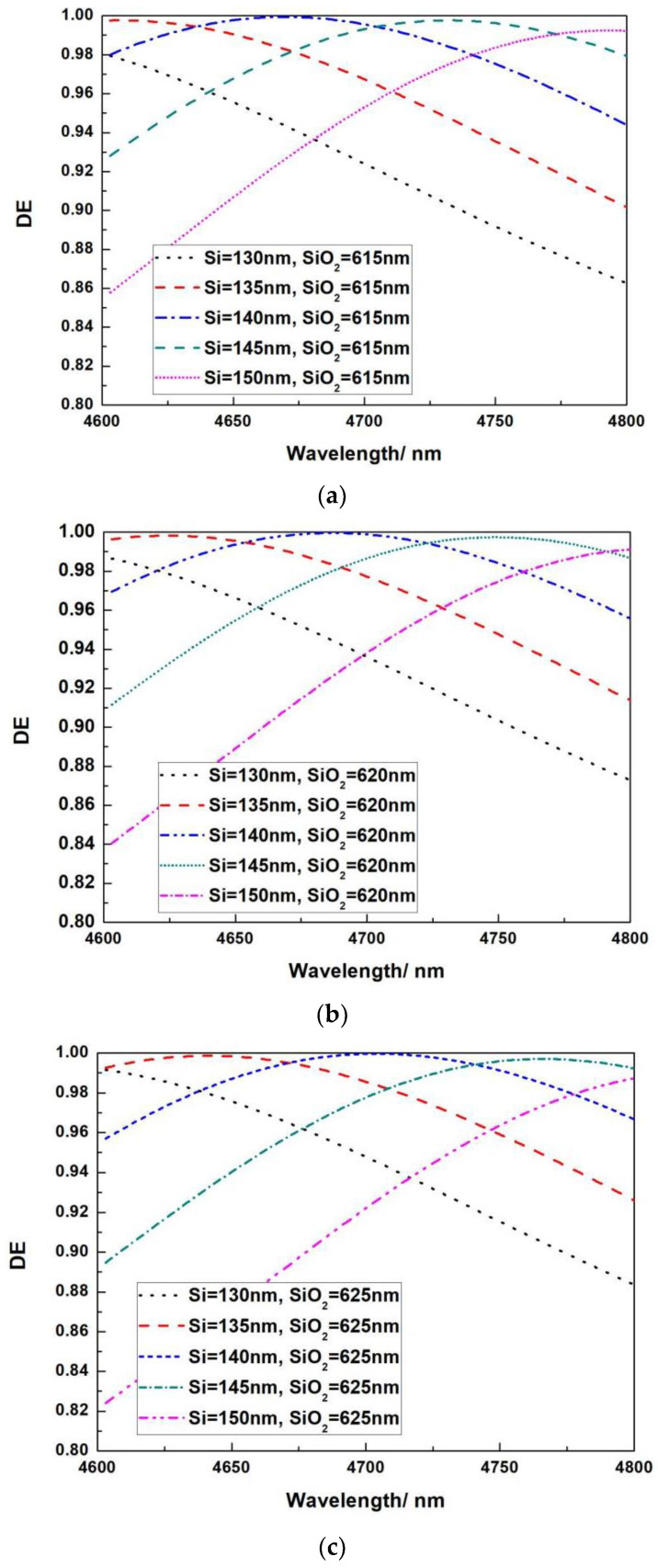
Fixed angle, wavelength, and diffraction efficiency. (**a**) Si with different thicknesses under 0.615 μm SiO_2_; (**b**) Si with different thicknesses under 0.620 μm SiO_2_; (**c**) Si with different thicknesses under 0.625 μm SiO_2_; (**d**) Si with different thicknesses under 0.630 μm SiO_2_; (**e**) Si with different thicknesses under 0.635 μm SiO_2_.

**Figure 6 micromachines-13-01706-f006:**
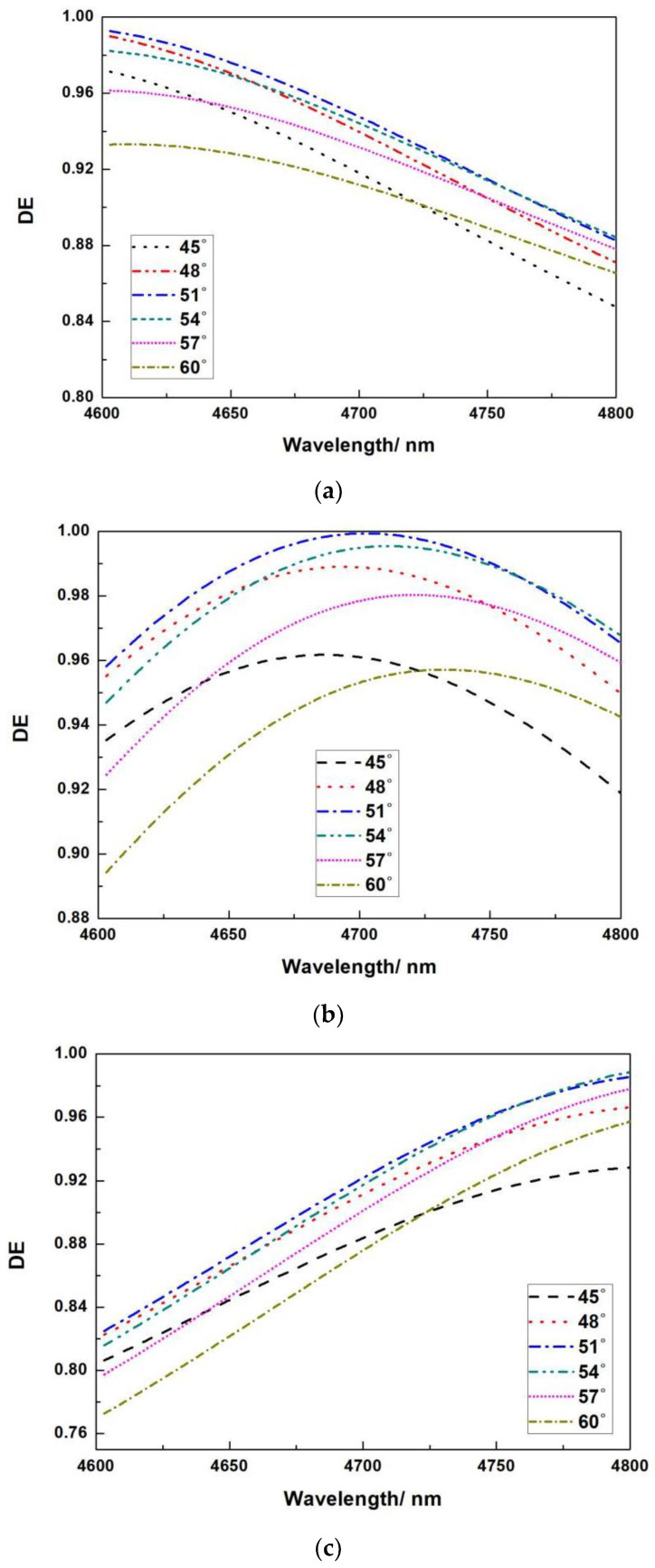
Relationship between wavelength and diffraction efficiency at different angles. (**a**) When the thickness of SiO_2_ is 0.620 μm and the thickness of Si is 0.130 μm; (**b**) when the thickness of SiO_2_ is 0.620 μm and the thickness of Si is 0.140 μm; (**c**) when the thickness of SiO_2_ is 0.620 μm and the thickness of Si is 0.150 μm. Relationship between wavelength and diffraction efficiency at different angles.

**Figure 7 micromachines-13-01706-f007:**
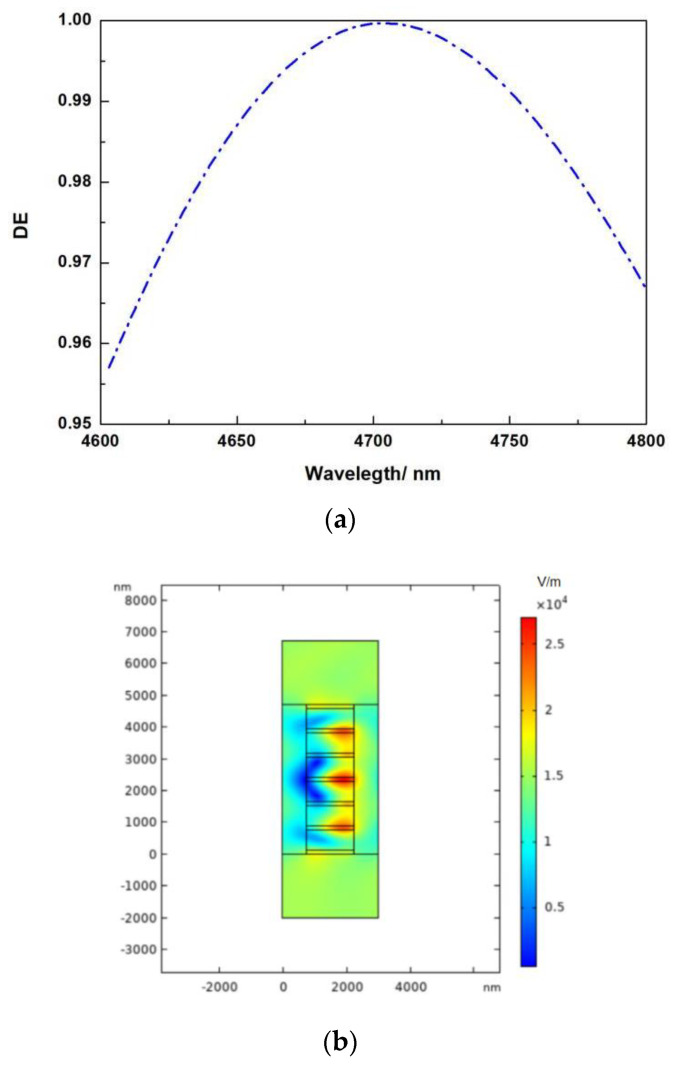
(**a**) The connection between incident wavelength and diffraction efficiency when the thickness of Si is 0.140 μm and the thickness of SiO_2_ is 0.620 μm. (**b**) The electric field distribution of the DBR structure in the transmission grating is shown when the incidence wavelength is 4.7 μm, the incident angle is 51.5667°, the thickness of Si is 0.140 μm, and the thickness of SiO_2_ is 0.620 μm.

**Figure 8 micromachines-13-01706-f008:**
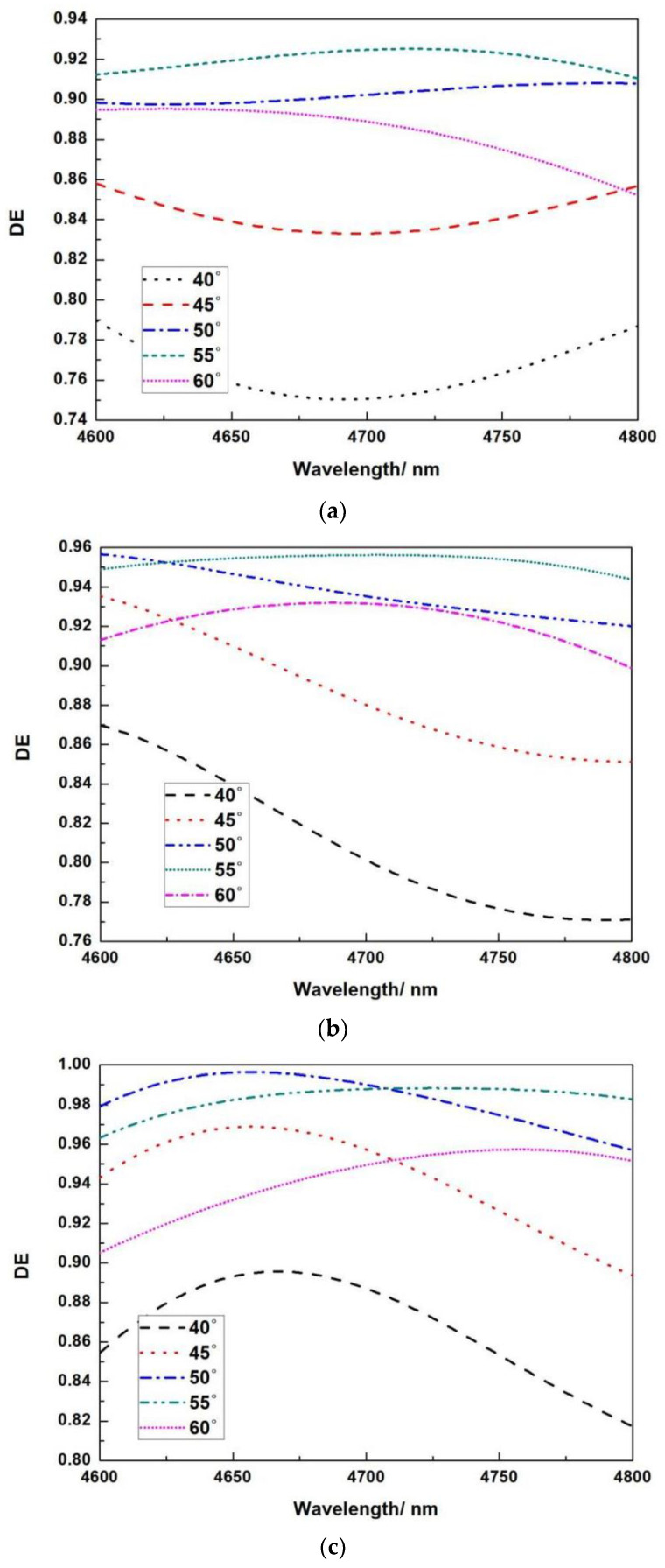
Relationship between wavelength and DE under different duty cycles and different incident angles, (**a**) 0.3, (**b**) 0.4, (**c**) 0.5 (**d**) 0.6, (**e**) 0.7. Relationship between wavelength and DE under different duty cycles and different incident angles.

**Figure 9 micromachines-13-01706-f009:**
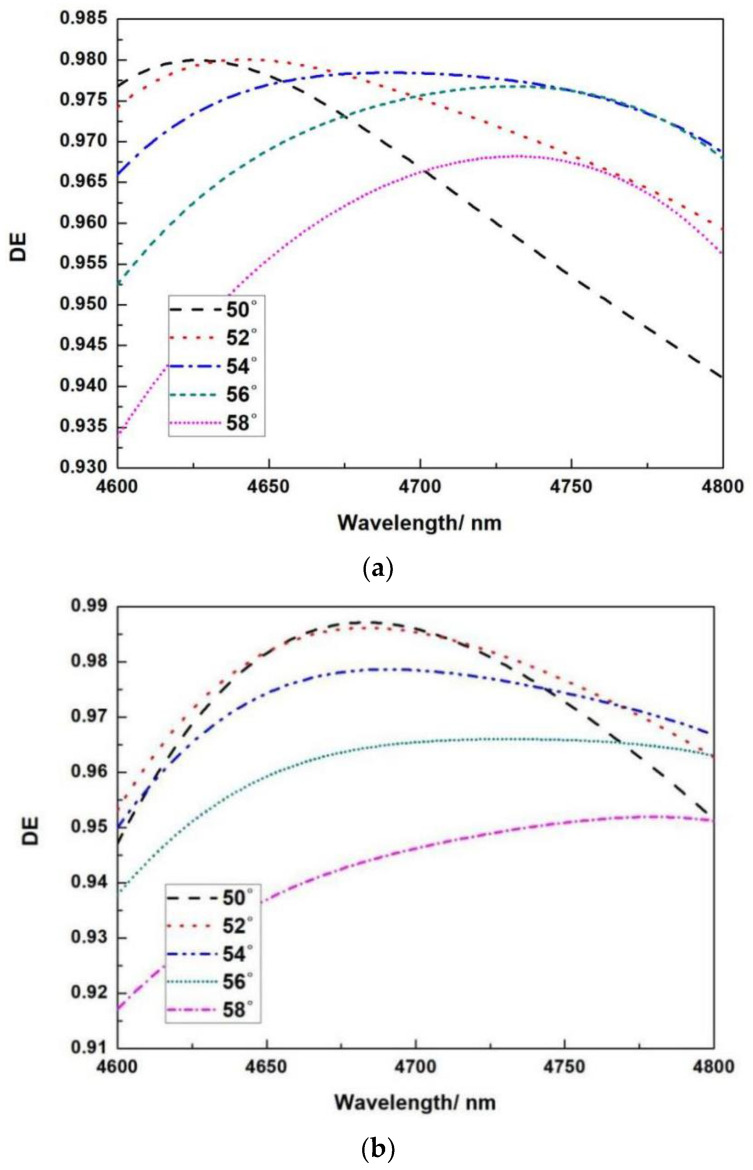
ff 0.45–0.55, the relationship between wavelength and DE at different angles. (**a**) ff = 0.45. the relationship between wavelength and DE at different angles. (**b**) ff = 0.55. the relationship between wavelength and DE at different angles.

**Figure 10 micromachines-13-01706-f010:**
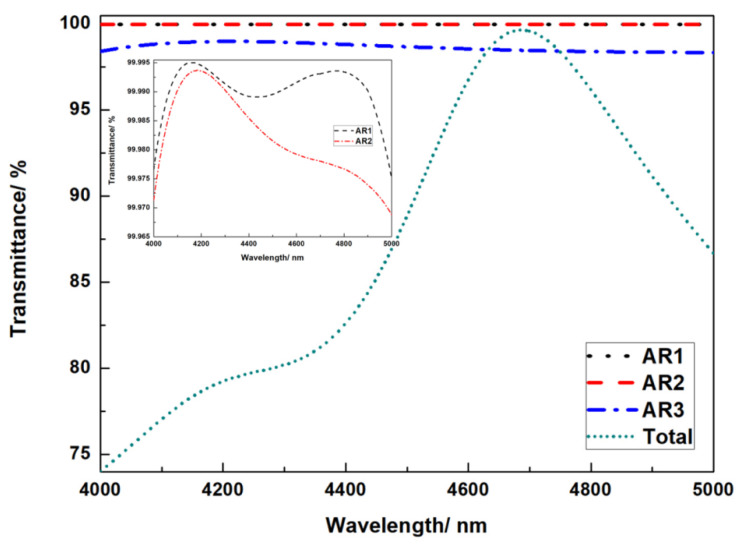
Spectral transmittance curve in mid-infrared band.

**Figure 11 micromachines-13-01706-f011:**
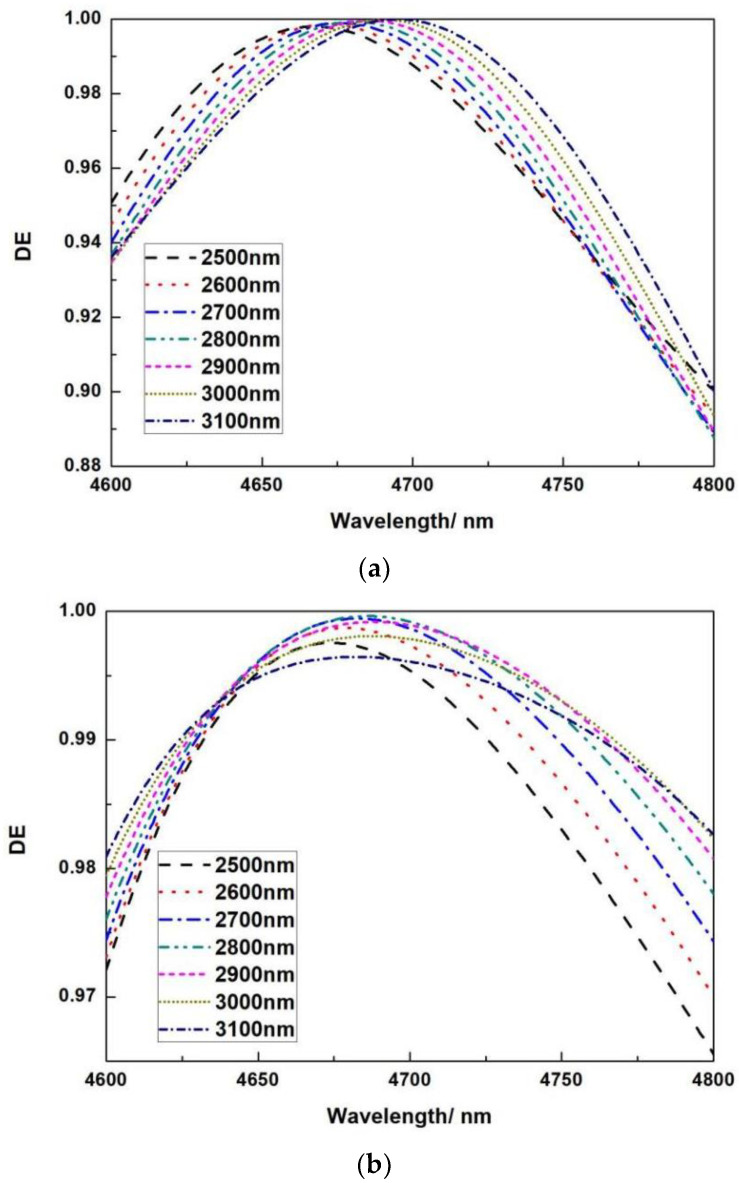
The relationship between wavelength and DE under different phase matching layer thicknesses; (**a**) phase matching SiO_2_-1 thickness; (**b**) phase matching SiO_2_-2 thickness.

**Figure 12 micromachines-13-01706-f012:**
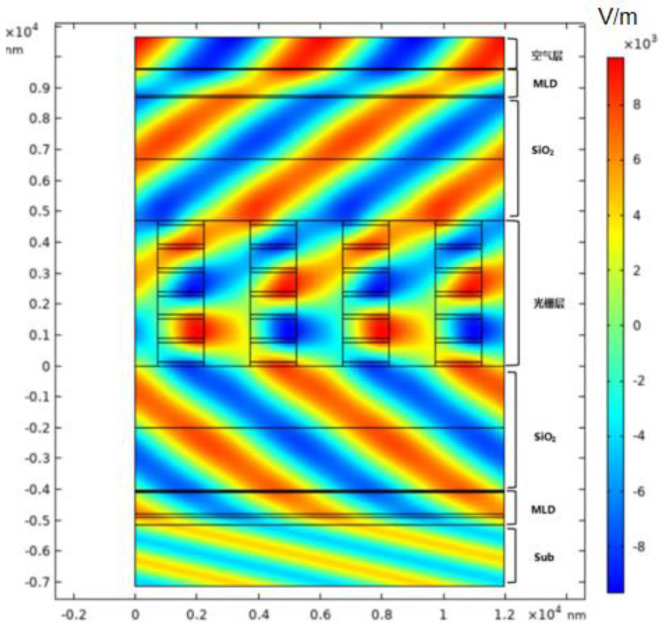
Total electric field intensity distribution.

**Table 1 micromachines-13-01706-t001:** Overview of multilayer dielectric transmission gratings in recent years.

Years	Units	Structures/Contributions	Performance
2010 [6]	Hongchao Cao, Changhe Zhou, et al.	Duty cycle 0.67, groove depth 1950 nm	λ = 800 nm, θ_0_ = 41.5°, DE > 92% (750~850 nm)
2011 [7]	Chun Zhou, Takashi Seki, et al.	Line density 1250 lines/mm, duty cycle 0.49, groove depth 1380 nm	θ_0_ = 30°, TE mode, λ = 800 nm, DE > 90% (750–850 nm)
2013 [8]	Bo Wang, Li Chen, et al.	Period 1100 nm, duty cycle 0.6,	λ = 1550 nm, DE > 95%
2020 [9]	Zhengkun Yin, Junjie Yu, et al.	Period 1624.6 nm, duty cycle 0.4	λ = 1550 nm, −2st Bragg DE > 95%
2021 [10]	Yongfang Xie, Wei Jia, et al.	Period 1270 nm, duty cycle 0.58	λ = 1550 nm, −1st Littrow DE > 95%, irrelevant loss 0.0013 dB

**Table 2 micromachines-13-01706-t002:** Transmission grating parameters to determine the phase matching layer of SiO_2_.

Scheme 4.	4.6–4.8 μm
Incidence angle	51.5667°
Substrate	Si
Period	3 μm
Filling Factor	0.5
Phase match SiO_2_-1	2.8 μm

**Table 3 micromachines-13-01706-t003:** Design results of the optimized AR coating system.

(a) AR1 coating system		
**AR1, Layer NO.**	**Layer material**	**Layer thickness/nm**
Incidence angle	13.3°	
Incident Medium	Si	
1	Ge	158.89 nm
2	SiO_2_	99.7 nm
3	Ge	857.71 nm
Exit Medium	Air	
(b) AR2 coating system		
**AR2, Layer NO.**	**Layer material**	**Layer thickness/nm**
Incidence angle	32.45°	
Incident Medium	SiO_2_	
1	SiO_2_	171.99 nm
2	Ge	120.96 nm
3	SiO_2_	500.27 nm
4	Ge	39.31 nm
5	SiO_2_	60.95 nm
Exit Medium	Si	
(c) AR3 coating system		
**AR3, Layer NO.**	**Layer material**	**Layer thickness/nm**
Incidence angle	51.5667°	
Incident Medium	Air	
1	Ge	41.5 nm
2	SiO_2_	677.24 nm
3	Ge	73.63 nm
4	SiO_2_	1244.34 nm
Exit Medium	SiO_2_

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
