# Peer review of "Optimal Design and Analysis of 4.7 μm Hybrid Deep Dielectric High Efficiency Transmission Gratings"

_micromachines, 2022, doi:10.3390/mi13101706_

Round 1

Reviewer 1 Report

In the paper “Optimal Design and Analysis of 4.7μm Hybrid Deep Dielectric High Efficiency Transmission Gratings” the authors describe the design and modelling of a transmission grating geometry for beam combination at a wavelength of 4.7 microns. Through modelling the authors show that a diffraction efficiency of 99.95% is possible for their structure through careful control of layer thickness, duty cycle, and the application of several anti-reflection coatings and matched layers.

Overall the authors make a reasonably in-depth study of their design. They investigate many contributing factors, as well as taking into account possible manufacturing errors and commenting on ways these devices may be realised. I believe the manuscript to be clear, though some additional information and descriptions may be good to see. The methods are described in detail and so should be simple to reproduce. The authors have previous publications on spectral beam combining lasers at various wavelengths, so it seems the next step will be to fabricate this structure to test with QCL’s. Therefore I believe this paper has novelty for publication, after some corrections.

Last line of the abstract does not make a lot of sense. I feel this needs to be rewritten to get the authors point across.

Section 2.1: The ‘Pure Si  grating’ shown here appears to be silicon encapsulated in SiO2. I do not see how this operates as a grating. Is this just a section of the model and the silicon is actually repeated several times to the sides to form a grating? If so this should be explained. Please also indicate in figure 2 which thickness parameter is changing to produce the results shown in figure 3.

Figure 5: Perhaps indication of the SiO2 layer thickness on each plot for ease of viewing. Same for figure 6.

Section 2.2.1 – the authors state that “…the pure Si grating, which has a maximum grating DE of 99.95%”. However, in section 2.1 they not for a pure Si grating the DE reaches a max of 89.9%. Is this an error in section 2.2.1?

Section 2.3 – The matrices (3) and (4) need to be reformatted.

Table 2 – For AR2 the exit medium should surely be Si and not air? Is this an error in the table or has this been modelled incorrectly as well?

Figure 10 – Perhaps plot a, b and c on the same axes with a legend? This would allow a better comparison between the coatings. Figure 11 could be incorporated as well.

Section 2.4 ‘these parts are arranged as shown in fig 12’ I don’t think that is correct fig ref, perhaps the authors meant figure 1?

Is the Si substrate necessary in this design? Would it be possible to fabricate this on SiO2 and do away with the sub-Si layer and AR1, replacing AR2 with a duplicate of AR3?

There appears to be an error in page numbering after page 9, with the digits not displaying correctly.

Author Response

Dear Reviewer,

Yours Sincerely,

Ye Wang

Reviewer 2 Report

In this manuscript, the authors have presented the optimal design and analysis of 4.7μm hybrid dielectric Gratings. The simulation results indicate that Littrow grating, hybrid multilayer dielectric grating with Si/SiO2 composite layers to achieve the high diffraction efficiency by using Finite Element Analysis method. However, the studied is classic and simple multilayer structure in mid-infrared domain. The similar structures were studied in many papers. The results of this manuscript are a little novelty, so it should be more clearly stated. The authors did not provide the theoretical or experimental results to agree with the simulation results. Besides, the written of this manuscript is pretty poor. I suggest this manuscript could not be considered at present style. So my decision is Rejected. Besides, the following issues should be addressed.

1. The abstract is in vague and unclear expression. The authors should carefully revised it to make your abstract more clear. Besides, the abbreviation DBR should be also indicated its full expression name when it is first shown up.

2. Please state the novelty in your manuscript compared to other similar works clearly. More specifically, what are effects and advantages of the simple structure? The authors should explanation their specific roles in structure in details.

3. More importantly, I strongly think that the total results and novelty of the manuscript are not enough as present style.

4. Besides, the written of this manuscript is pretty poor. There are too many format errors (such as Figure 6, Figure 8, Figure 11, Figure 12; there is no expression name corresponding to abbreviationff ...) and uneasy understand parts. The authors should carefully polish their paper.  

5. In this paper, the authors only presented the simulation results about DE in different parameters although they draw a lot of pictures. The total tasks is not so clear and not enough for a high quality of scientific work.

Author Response

Dear Reviewer,

Yours Sincerely,

Ye Wang

Author Response

Dear Reviewer,

Yours Sincerely,

Ye Wang

Round 2

Reviewer 2 Report

The revised manuscript have been great improved.